# Inverted battery design as ion generator for interfacing with biosystems

Chengwei Wang[1], Kun (Kelvin) Fu[1], Jiaqi Dai[1], Steven D. Lacey[1], Yonggang Yao[1], Glenn Pastel[1], Lisha Xu[1], Jianhua Zhang[2] & Liangbing Hu[1]

In a lithium-ion battery, electrons are released from the anode and go through an external electronic circuit to power devices, while ions simultaneously transfer through internal ionic media to meet with electrons at the cathode. Inspired by the fundamental electrochemistry of the lithium-ion battery, we envision a cell that can generate a current of ions instead of electrons, so that ions can be used for potential applications in biosystems. Based on this concept, we report an 'electron battery' configuration in which ions travel through an external circuit to interact with the intended biosystem whereas electrons are transported internally. As a proof-of-concept, we demonstrate the application of the electron battery by stimulating a monolayer of cultured cells, which fluoresces a calcium ion wave at a controlled ionic current. Electron batteries with the capability to generate a tunable ionic current could pave the way towards precise ion-system control in a broad range of biological applications.

[1] Department of Materials Science and Engineering, University of Maryland College Park, College Park, Maryland 20742, USA. [2] Metabolic Diseases Branch, National Institute of Diabetes and Digestive and Kidney Diseases, National Institutes of Health, Bethesda, Maryland 20892, USA. Correspondence and requests for materials should be addressed to L.H. (email: binghu@umd.edu).

All living creatures including human beings, animals, plants and microbes contain ionic systems, in which most of their biological activities involve ion transport. The human brain responds to stimuli sensed from the peripheral systems and transmits responses back through the nervous system. During this process, ions act as charge carriers and are involved in each step. Among these biological processes, their energy is supplied by adenosine triphosphate (ATP) molecules, which exist as complex ions as well[1]. Ion transfer is therefore critical for vital biological functions but has yet to be controlled by an inverse battery design as demonstrated in this work.

On the other hand, advanced computers and robots are becoming better at simulating and mimicking biological functions. In robots, information is processed inside the central operating unit, and then electrical signals are sent out to control the motion of the robots which have electric motors for artificial muscles. In these electronic systems, batteries, especially Li-ion batteries, supply the electrical energy to drive the movement of electrons. However, since electrons and ions travel in different conductive media, an electronic system cannot directly interact with an ionic system. Otherwise, disruptive electrochemical reactions would occur when electrons and ions cross the electrode/electrolyte interface, which would harm the whole system. In traditional batteries, the electrons go through external electrical cables to interact with electronic devices, and the ions transfer through internal ionically conducting media. When a battery powers an ionic system through electrical cables, there is no current flow before the voltage reaches the potential to initiate the electrochemical reaction. To better interact with ionic systems, an ideal strategy is to design an ionic device to directly 'communicate' with ions in ionic systems.

Most recent attempts are using indirect methods to communicate with ion system through some intermedia. For example, Suo and his coworkers recently developed a stretchable, transparent, ionic conductor, which can transmit alternating electrical signals with the help of an electromagnetic field by forming an electrical double layer at the electrode/electrolyte interface[2,3]. Other methods use an electrochemical reference electrode as a bridge to initiate interactions between ionic and electronic systems. Based on this design, the patch clamp technique was developed to study ionic channels in cells by Erwin Neher and Bert Sakmann, who were awarded the Nobel Prize in Physiology or Medicine in 1991 for their work[4–6]. This method has also been used to study the ionic transport properties of synthetic ionic channels[7–10]. Although the aforementioned methods enabled an electronic system to interact with an ionic system, as far as we know, there has not yet been an effective ionic device that can directly communicate with ionic systems. Since the dominating charge species in biosystems are ions, if a continuous ionic current can be supplied, we can directly measure the ionic processes in a biosystem and even supply energy to the biosystem. Therefore, an ionic device that can generate ionic signals to directly interact with ionic systems is of great interest.

Here, for the first time to the best of knowledge, we demonstrate a concept of the electron battery by inverting the design of the traditional battery, which can generate an ionic current instead of an electric current. Since the output charge carriers of the electron battery are ions, when the electrodes of the electron battery connect with an ionic system through ionic cables, the ions can enter or exit the ionic system freely without a threshold voltage for an electrochemical reaction. For this case, the interface between the electrodes and the ionic system consists of two ionic mediums which allow the continuous transport of ions. Therefore, the applied ionic current can be linearly adjusted from zero by fine tuning the voltage, which is especially important for biosystems in which the voltage and current involved in biological activities are low. With these unique ionic properties, the proposed electron battery can be a strategy to solve biological challenges.

## Results

**Electron battery for potential biological applications**. For potential applications in the human body, Fig. 1 depicts the systems in which the electron battery can potentially interact. Most biological activities in the human body involve the transport of ions through ion channels located on the cellular membrane of each living cell. For example, when a signal is transmitted along the axon of a neuron in the human body, $Na^+$ ions move into the neuron cell, and $K^+$ ions diffuse out of the cell through voltage-gated channels on the membrane. In this process, the membrane potential flips from negative to positive[11]. After the signal passes through, the sodium-potassium pump moves $Na^+$ ions out and $K^+$ ions in to reset the membrane potential to negative. Similar ionic processes happen in muscle cells as they contract and relax, in which $Ca^{2+}$ and other ions flow into or out of the cells. As shown in Fig. 1, by using different electrode materials and ionic cables soaked with various ions, the electron battery can selectively generate ion flow of $Li^+$, $Na^+$, $K^+$, $Ca^+$ and so on, which are typical ions in many biological processes inside the body. Therefore, the tunable ionic current from the electron battery could potentially change the ion concentrations/charges outside of the cell membrane, resulting in direct alteration of the membrane potential to modulate cellular behaviour. The electron battery can interact with the cells at any current level since it does not have a threshold voltage.

Moreover, since no electrochemical reactions happen in the biosystem during the interaction, the electron battery can directly apply electrochemical energy to the biosystems without causing decomposition of the electrolyte and tissues. If integrated with micro-electronic devices, the electron battery can be designed for implantable biomedical devices with various functions, such as drug delivery and activation for specific treatments. As an example, lithium salt is one of the most effective treatment methods for bipolar disorder[12–14]. However, lithium can also cause side-effects especially at high dosages[13,15]. If an implantable electron battery with a lithium anode acted as the $Li^+$ source, lithium could be controllably released as needed, which could be a more effective and safer treatment. Therefore, we envision that the electron battery can potentially provide new strategies to stimulate and interact with biosystems and lead to novel strategies to treat many human diseases with biomedical applications.

**Intrinsic differences in traditional and electron batteries**. Figure 2a compares the working mechanism of both traditional and electron batteries. A traditional battery includes the cathode and anode, which are connected by an ionic medium (for example, liquid electrolyte); externally, the electrons travel through a circuit to power devices. Oppositely, the inside of an electron battery is connected by an electrical conductor that only allows electrons to be transported from the negative to the positive electrode, while the ions flow through ionic cables and are transported externally to create a circuit of ions. When these two types of batteries interact with an ionic system where the charge carriers are cations and anions, their electrochemical behaviours significantly differ. The charge carriers in an electrical system are electrons and holes, which cannot be transported in an ionic system. When the positive and negative electrodes of a traditional battery connect with an ionic system through electrical cables, there will be no current flow without the occurrence of electrochemical reactions. Since each electrochemical reaction has an onset voltage ($V_E$), before the applied voltage reaches the onset

voltage, the current is close to zero (Fig. 2b). When the voltage reaches or exceeds the onset voltage, the redox species will be reduced at the negative electrode and be oxidized at the positive electrode. During the electrochemical reaction, electrons exchange at the corresponding electrode–electrolyte interfaces and cause the current to increase (Fig. 2b). However, the electrochemical reactions can make the ionic system unstable, which is especially fatal for a living biosystem where the electrochemical reactions can cause the decomposition of cells, tissue or water.

Considering that most ionic processes in biosystems involve both low currents and voltages, the electrical power from the conventional battery cannot act as a source for the ionic system. When an electron battery connects with an ionic system through ionic cables, the ionic charge carriers will drift along the electrical field. Since both the ionic cable and ionic system can transport ions, the ionic charge carriers can cross the electrode–electrolyte interface and travel into the ionic system freely (Fig. 2a). For this case, there is no threshold voltage for the electrochemical reaction, therefore, the ionic current will follow Ohm's Law, where the current is linear with the applied voltage. The slope of the I–V curve is determined by the ionic resistance of the system (Fig. 2b). The linearity will allow for fine control and tuning of the ionic current by adjusting the voltage from zero instead of $V_E$ for traditional batteries. Even with a small voltage bias, a non-zero ionic current can be achieved (Fig. 2b), which is especially important for applications in biosystems where the living cells and tissues are delicate.

**Grass ionic cable**. Similar to the electrical cables that conduct electrons, ionic cables facilitate the transport of ions. Many materials, such as metals, carbon and doped semiconductors are excellent electrical conductors. However, ionically conductive materials are limited and their conductivities are much smaller than electrical conductors. Among the most common ionic conductors (that is, conductive polymers, ionically conductive ceramics, ionic liquids and salt solutions), liquid phase salt solutions typically have the highest conductivity, around 10–700 mS cm$^{-1}$ at room temperature[16,17]. However, the solutions have no mechanical strength to form continuous stable shapes.

Here we developed a flexible ionic cable composed of a natural grass stem soaked in an aqueous salt solution. To create a cable-like conductor, mechanically robust materials, such as a cotton string or grass, are required to contain the liquid electrolyte. Note that a cotton string and natural grass are structurally different and thus, liquid electrolyte is stored in distinct ways. The soaked cotton string stores the electrolyte around the material (Supplementary Fig. 1a), while the natural grass possesses vertically aligned microchannels that can hold the electrolyte (Supplementary Fig. 1a,b). Therefore, the natural grass retains the liquid electrolyte while maintaining flexibility, which is advantageous for ionic cables. The grass stem is from the *Poa pratensis*, a common lawn grass on the University of Maryland campus. The inset of Fig. 3a displays the 30–50 cm long grass stem with a relatively uniform diameter of 1–2 mm, which can be directly used to fabricate ionic cables after the leaves are removed. The ionic cables were fabricated by soaking the grass stems into a saturated LiNO$_3$ aqueous solution under vacuum to enhance infiltration. The surface of the ion soaked grass stem was then covered by a thin thermal shrink tube to prevent the solution from evaporating. After the fabrication process, the ionic cable

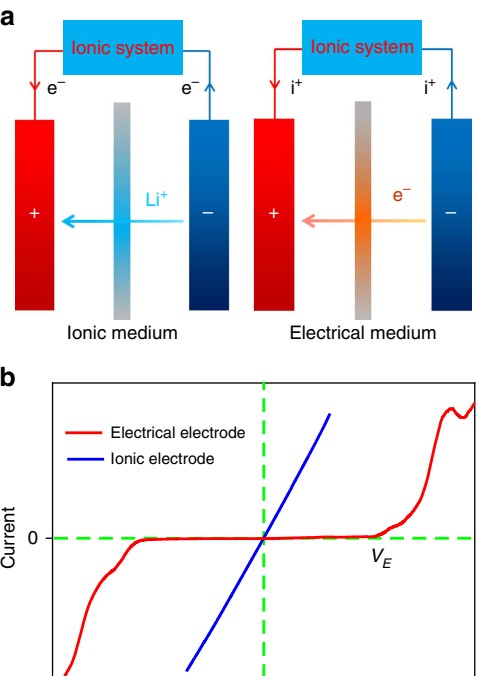

**Figure 2 | The intrinsic differences between the traditional battery and the electron battery when interacting with an ionic system.** (**a**) Schematics of a traditional battery (left) and an electron battery (right). For the traditional battery, the anode and cathode are separated by an ionic medium that allows ions to diffuse while electrons travel via an external circuit. For the electron battery, an electrical medium is used to conduct electrons internally and the ions are transported through the external circuit. (**b**) Illustration of I–V curves where a traditional battery and an electron battery interact with an ionic system. The curve of the electrical electrode exhibits an onset voltage ($V_E$), where the electrochemical reaction occurs. The ionic electrode shows a different behaviour (following Ohm's Law) where the current changes linearly with the potential.

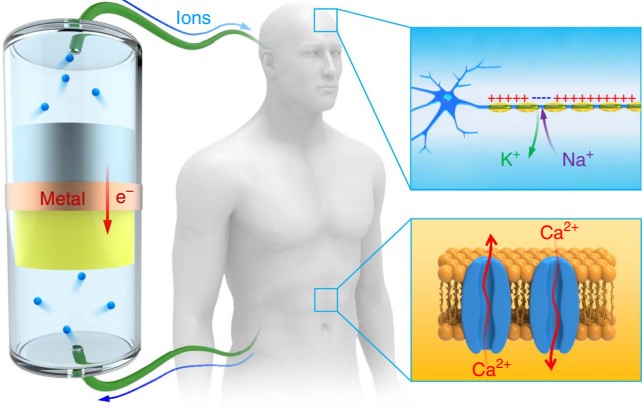

**Figure 1 | Schematic of an electron battery for potential biological applications.** In the human body, ions govern vital biological functions, such as signal transmittance in neurons, the movement of muscles and the activities of other cells. An electron battery composed of metal anodes and corresponding high voltage cathodes can continuously generate ions and power the motion of ions in various biosystems. Internally, the electron battery is connected by an electrical conductor that allows the transport of electrons, and its external electrodes are connected with the human body through ionic cables.

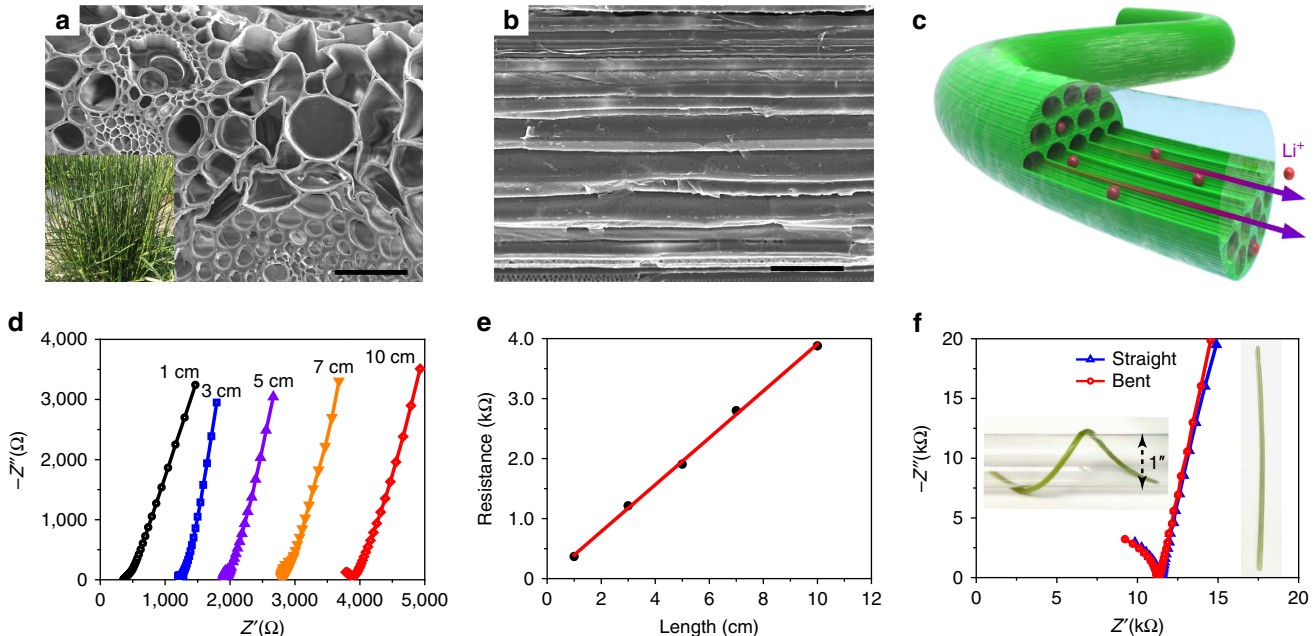

**Figure 3 | Characterization of the grass ionic cable.** SEM images of the (**a**) cross section and (**b**) longitudinal-section of a grass stem. Scale bar, 50 μm. The inset is a digital image of the long grass stems. (**c**) Schematic of the ionic cable made by soaking a grass stem in an aqueous lithium salt solution. The lumens of grass provide continuous channels for the transport of the ions. (**d**) EIS spectra and (**e**) the corresponding resistances of the grass ionic cables with similar diameters (∼2 mm) and different lengths (1–10 cm), which indicates that resistance has a linear relationship with length. (**f**) EIS spectra of a 10 cm grass ionic cable before and after being bent around a 1″ glass tube. Insets correspond to bent and straight grass ionic cables, respectively.

maintains the original grass' features. Figure 3a,b exhibit scanning electron microscopy (SEM) images of the cross section and longitudinal views of a grass stem. The stem of natural grass is full of micro-vessels, which are vertically aligned across the entire blade of grass. The size of the micro-vessels varies from several micrometres to several tens of micrometres (Fig. 3a,b), which is suitable for absorbing and containing the electrolyte solution due to capillary effects. When these longitudinally continuous vessels are filled with the aqueous salt solution, they act as an ionic cable. The aligned structure facilitates guided transport of ions along the longitudinal direction (Fig. 3c). Figure 3d shows Nyquist plots used to determine the conductivity of the ionic cables at different lengths. The intercept of each curve with the real axis is used to estimate the resistance of the ionic cable. According to Fig. 3e, the resistance increases almost linearly with the length of the ionic cable, indicating a relatively constant ionic conductivity. The conductivity of the ionic cable (∼80 mS cm$^{-1}$) is then calculated based on the slope of the resistance-length curve. This is comparable to a pure LiNO₃ aqueous solution (∼150 mS cm$^{-1}$)[18] and almost one order of magnitude higher than organic-based liquid electrolytes[17,19]. Due to the excellent mechanical strength and flexibility, the grass ionic cable can be bent into random shapes in a similar manner as a conventional electrical cable, yet, its ionic conductivity is nearly constant (Fig. 3f). The stable ionic conductivity can be attributed to the structure's long continuous micro-channels, which retain the liquid electrolyte when bent.

**Electron battery**. The electron battery is composed of a lithium metal anode and a vanadium oxide ($V_2O_5$) cathode (Fig. 4a). Both the anode and the cathode are sealed in glass tubes filled with organic electrolyte. One end of the glass tube is sealed with the ion exchange membrane (IEM) which retains the electrolyte and allows ions to pass through (Supplementary Fig. 2; Supplementary Methods section for IEM selection rules). The electrodes are connected with an electrically conductive wire that allows the

transport of electrons and serves as the internal circuit. Two LiNO₃ soaked grass ionic cables are attached onto the IEM of each electrode to conduct ions and serve as the external circuit. To achieve further enhancements in performance, the cathode consists of $V_2O_5$ nanowires synthesized by a previously reported hydrothermal method[20]. The diameters of the $V_2O_5$ nanowires are around several tens of nanometres, while their lengths are several micrometres (SEM image in Supplementary Fig. 3). The high aspect ratio allows the nanowires to form a network structure, which can decrease the distance of lithium-ion diffusion and facilitate the transport of electrons. To characterize the electrochemical performance of the electron batteries, multiple discharge voltage profiles were taken at different ionic loads. Unlike the conventional battery set-up, the voltage profile of an electron battery cannot be recorded from its anode and cathode directly, since they have already been electrically shorted by the electrical conductor. Instead, a lithium metal reference electrode was placed close to the cathode to record the voltage profile between them. The inset of Fig. 4b depicts the electrochemical impedance spectroscopy (EIS) spectra of two electron batteries with different ionic cables. Their ionic resistances are calculated to ∼30 kΩ and 135 kΩ, respectively. The voltage profiles of these two batteries are shown in Fig. 4b, where the battery with the lower resistance discharges much faster than the one with the higher resistance. Thus, the electron battery exhibits similar electrochemical behaviour to a traditional battery. Since the battery with the higher resistance discharges at a slower rate, the voltage profile displays clear plateaus at ∼2 V and ∼3 V, which corresponds to $V_2O_5$ (refs 21–23).

To demonstrate the interaction with an ionic system, the electron battery was used to drive the migration of ions in the ionic system. To visually see the process, blue coloured copper ions were selected to demonstrate the migration along a LiNO₃ solution soaked cotton string (∼1 mm thick, 8 cm long; Fig. 4c). When the electron battery is connected to the cotton string by two grass ionic cables, the lithium ions will diffuse from the anode to the cathode through the external ionic circuit to form an ionic

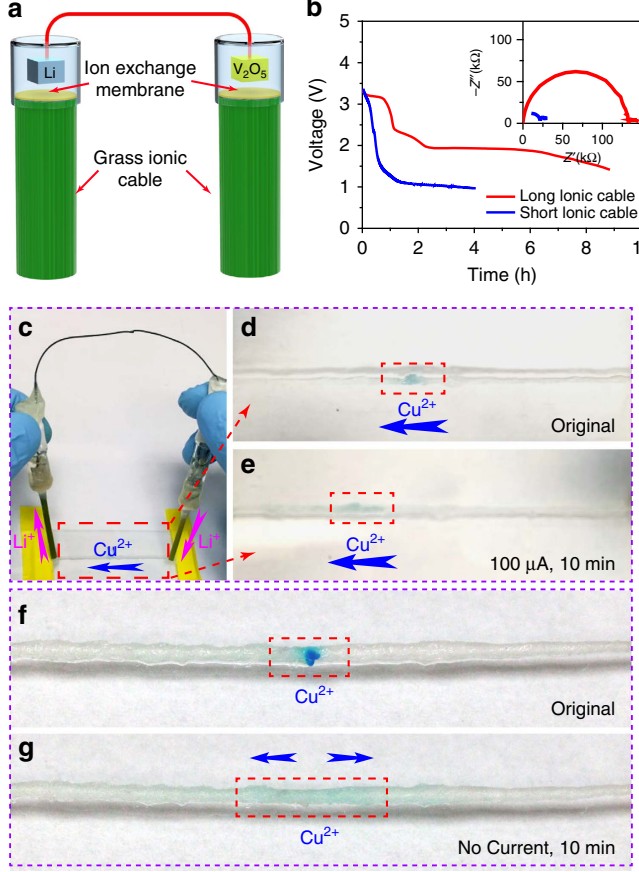

**Figure 4 | Characterization of the electron battery.** (**a**) Schematic of the electron battery with a lithium metal anode, $V_2O_5$ nanowire cathode and grass ionic cables. (**b**) Discharge profiles of electron batteries controlled by ionic cables with different ionic resistances. The inset is the EIS spectra of the corresponding electron batteries. (**c**) A digital image of the electron battery demonstration to drive blue copper ions along a $LiNO_3$ solution soaked cotton string ($\sim1$ mm thick and 8 cm long). Digital images of (**d**) the initial and (**e**) moved state of blue copper ions before and after being driven by the electron battery. This indicates that the electron battery can cause ions to migrate in an ionic system. Digital images of (**f**) the initial and (**g**) diffused states for the control experiments. In the control experiments, no current was applied and the blue copper ions randomly diffused along the cotton string in both directions.

current. The flow of $Li^+$ will create an electrical field, where the blue coloured copper ions will drift towards the cathode. Initially, the blue coloured ions were placed at the centre of the string, which is marked as shown in Fig. 4d. After the electron battery is connected to the two ends of the string and discharged at a current of $\sim100\,\mu A$ for 10 min, the blue coloured ions diffused towards the cathode side of the battery (Fig. 4e). In the control experiment, when no current was applied to the string, the blue ions randomly diffused along the string from the centre (Fig. 4f) towards the ends in both directions (Fig. 4g). This behaviour is drastically different from the electron battery's driven process. This simple yet straightforward method demonstrates that the electron battery can directly interact with the ionic system without causing electrochemical reactions, which significantly differs from traditional batteries.

**Electron battery for biological applications**. As a proof-of-concept, we selected cultured living cells as a simple biosystem to demonstrate the interaction between the electron battery and a

biosystem. Here, the electron battery was used to stimulate the movement of calcium ions ($Ca^{2+}$) inside the living cells by generating a continuous ionic current. Figure 5a is a schematic of the stimulation experiment, where the electron battery applies an ionic current to the cells through the ionically conductive cables and tips. To make the ions compatible with the living biosystem, the grass ionic cables were soaked with NaCl solution instead of a lithium salt solution. Small electrode tips were specifically constructed using syringe needles to enhance the contact between the cells. Two syringe needles (inner diameter, $\sim0.5$ mm) filled with 150 mM NaCl in a 1% agarose hydrogel were attached to the grass ionic cable to ensure reliable contact with the cells. The electrode materials are sealed in a lithium organic electrolyte-filled glass tube by an IEM and thus, ion exchange occurs at the interfaces between the electrodes and the grass ionic cables.

The ionic current of the drifted $Na^+$ ions will generate a continuous electric field to stimulate the cells. When the cells are stimulated, the calcium ions will be released from the endoplasmic reticulum, the intracellular calcium storage pool, through $Ca^{2+}$ channel[24]. Meanwhile, some messengers will diffuse from the stimulated cell to its neighbouring cells through gap junctions. These messengers will cause the internal $Ca^{2+}$ release process to occur in neighbouring cells. As shown in Fig. 5a, a calcium indicator, Fluro-4, exhibits enhanced green fluorescence on $Ca^{2+}$ binding. Note that this indicator is used to identify the $Ca^{2+}$ concentration in the living cells. Figure 5b shows the fluorescence intensity changes of the calcium indicator in the living HEK293 cells after stimulation from the electron battery using $\sim30\,\mu A$ of ionic current. The initial fluorescent area at 0 min (Fig. 5b,c) was due to mechanical stimulation from the electrode's needle tip breaking the monolayer of cultured cells[25–27]. After the ionic current was applied to the cells, the electrical stimulation caused the release and accumulation of calcium ions within the living cells. The binding of calcium ions to the fluorescent indicator leads to an increase in the fluorescent intensity. Moreover, due to the interaction between cells, the fluorescence intensity spreads along the cells to form a wave-like fluorescent image. After 30 min of continuous stimulation from the electron battery, the fluorescence signal reaches nearly all areas surrounding the living cells (Fig. 5b).

To further demonstrate that the fluorescence intensity changes were due to the ionic current, a control sample was stimulated by the same electrode tips from the electron battery without an applied ionic current (Fig. 5c). Similar with the sample showed in Fig. 5b, the cells showed initial fluorescence around the tip-touched areas due to mechanical stimulation. However, without further stimulation from the ionic current, the mechanical stimulated fluorescence became dim over time instead of spreading out or getting brighter, which agrees with the literature[26,27].

Therefore, the ionic current of the electron battery can successfully interact with living cells to stimulate the $Ca^{2+}$ wave. Moreover, during the stimulation process, the ionic current from the electron battery did not cause electrochemical reactions in the biosystem. If a traditional battery were used to stimulate the cells, the water within the biosystem would undergo electrolysis to generate a continuous ionic current, which would harm the living cells.

## Discussion

In summary, by inverting the traditional battery design, we for the first time to the best of our knowledge, demonstrate an electron battery system that can generate an ionic current. When the electron battery interacts with an ionic system, there is no

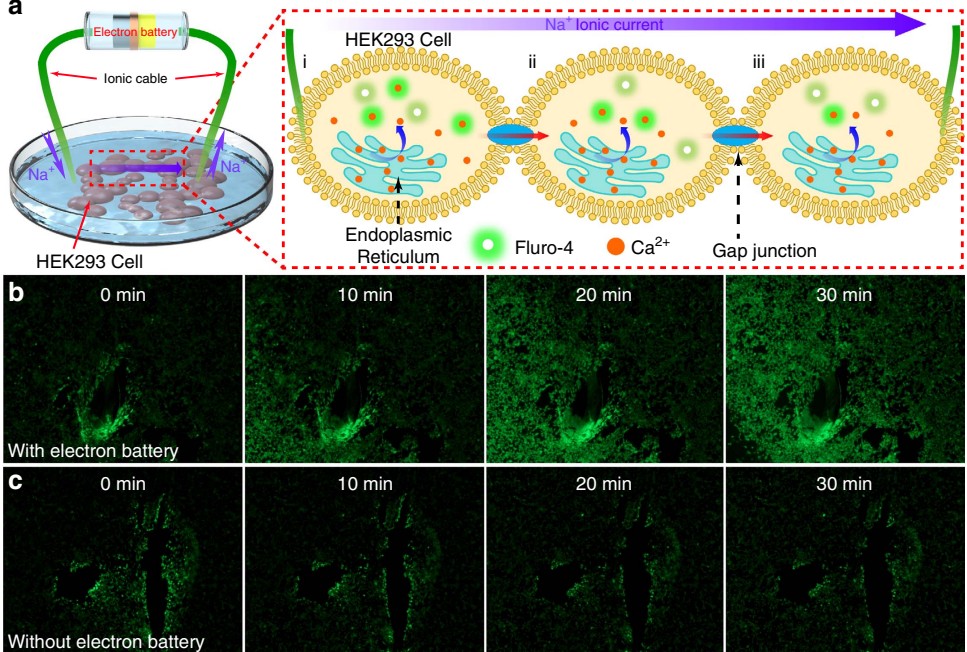

**Figure 5 | Demonstration of the electron battery in the biosystem.** (**a**) A schematic of the experimental set-up and the mechanism for calcium wave propagation through HEK293 living cells when stimulated by ionic current from the electron battery. The green fluorescent images of calcium waves produced in HEK293 cells (**b**) with and (**c**) without ionic current stimulation at different time intervals. The fluorescent areas at 0 min are the initial calcium waves mechanically stimulated by the tips of the ionic cables.

threshold voltage where electrochemical reactions, which means that the applied voltage or current can be finely tuned from zero. A new component of the electron battery system, the ionic cable, was developed using a natural grass stem. The grass stem possesses vertically aligned microchannels that retain the liquid electrolyte and allow for fast ionic transport along the channels. The conductivity of the grass ionic cable is $\sim 80\,\mathrm{mS\,cm^{-1}}$, which is almost one order of magnitude higher than organic-based liquid electrolytes. The electron battery successfully stimulated a $Ca^{2+}$ wave in the cultured living cells, which demonstrates this novel battery design for biological applications. Note that $Ca^{2+}$ flows in and out of muscle cells to stimulate muscle movement. Most biological processes involve the transport of ions within living organisms thus, the proposed electron battery can potentially interact with these ionic processes and be applied to additional biological applications. Therefore, we envision the electron battery can offer new strategies for electro-biomedical applications, such as the treatment of nerve system damage, heart disease (by altering contraction), Alzheimer's/Parkinson's disease (devices to control neuron stimulation), muscle stimulation as well as novel biosensors that can monitor human health.

## Methods

**Fabrication of electron battery.** The cathode material, vanadium oxide ($V_2O_5$) nanowires, was synthesized using a previously reported method[20]. Briefly, 0.8 g of $V_2O_5$ powder ($\geq 99.6\%$, Sigma-Aldrich) was added to 60 ml of DI water and stirred for 1 h. Then 10 ml of 30% hydrogen peroxide ($H_2O_2$) was added to the solution and continuously stirred for 2 h. Afterwards, the solution was heated at 210 °C for 100 h in an autoclave to facilitate the hydrothermal reaction. The final $V_2O_5$ nanowires were washed with DI water several times and vacuum filtrated into a film. The $V_2O_5$ nanowire film was cut into small pieces and wrapped in a conductive carbon cloth to serve as the cathode. Both the cathode and the lithium anode were sealed in a glass tube filled with 1 M $LiPF_6$ in ethylene carbonate/ diethyl carbonate (1:1 vol%) electrolyte. One end of the glass tube was sealed with a cation exchange membrane (CMI-7000S Cation Exchange Membrane, Membranes International Inc.) to allow the access of ions. The grass ionic cable originates from the stem of *Poa pratensis* grass. Specifically, the grass stems were soaked into the $LiNO_3$ (Sigma-Aldrich) saturated aqueous solution under vacuum. The ion soaked grass was then covered by a thin thermal shrink tube to prevent the electrolyte

from evaporating. Afterwards, the ionic cables were attached to the cation exchange membrane sealed glass tube electrodes using ion soaked cotton connectors which conduct ions.

**Preparation of bio-samples.** HEK293, a cell line derived from human embryonic kidney cells, were purchased from ATCC (Cat# CRL-1573). Fluro-4 direct, a calcium indicator, was purchased from ThermoFisher Scientific (Fluo-4 Direct Calcium Assay Kit; Cat# F10471). A high glucose DMEM cell culture medium (Cat# 11995065) and fetal bovine serum (Cat# 16000044) were purchased from ThermoFisher Scientific as well.

**Bio-testing methods.** One day before the experiment, HEK293 cells were split into 6-well cell culture plates at 90% confluency in a 2 ml cell growth medium (DMEM medium supplied with 10% fetal bovine serum) per well. The cells were cultured overnight in an incubator with 5% $CO_2$ at a constant temperature of 37 °C. To load the calcium indicator into the cells, in each well of the plate, 1 ml of the cell growth medium was replaced with 1 ml of 2 × working solution of the fluro-4 calcium indicator prepared according to manufacturer's instructions. The cells were incubated at the same conditions for an hour and tested with the electron battery. Two syringe needles (inner diameter, $\sim 0.5\,\mathrm{mm}$) filled with 150 mM NaCl in a 1% agarose (Sigma-Aldrich) hydrogel were attached to the grass ionic cable and acted as small electrode tips to ensure good contact with the cells. During the stimulation, the ionic current was about $\sim 30\,\mu A$. On stimulation, the green fluorescent calcium waves produced within cells were visualized under a fluorescent microscope (Nikon TS100).

**Materials characterization.** The morphologies of the grass stem and $V_2O_5$ nanowires were conducted with a Hitachi SU-70 FEG-SEM at 10 kV.

**Electrochemical measurement.** EIS was conducted with a BioLogic VMP3 potentiostat in the frequency range of 500 mHz to 1 MHz with a 50 mV AC amplitude. To measure the EIS of the grass ionic cables, two titanium metal strips were attached to the ends of the grass cable, where the $LiNO_3$ soaked cotton was used to wrap the metal to the end of the grass cables to achieve ionic connections. The voltage profiles of the electron battery were recorded between the cathode and the lithium metal reference electrode located near the cathode. Note that the anode and cathode were connected by an electrical cable. In the experiments, a Keithley 2,400 SourceMeter was connected to the circuit to measure the current.

**Data availability.** The data that support the findings of this study are available from the corresponding author upon request.

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

## Acknowledgements

We acknowledge the support of the Maryland NanoCenter and its NispLab. This work was supported as part of The Center for Research in Extreme Batteries (CREB) funded by University of Maryland College Park (UMD), Army Research Lab (ARL) and The National Institute of Standards and Technology (NIST). This work was also supported as part of the Nanostructures for Electrical Energy Storage (NEES), an Energy Frontier Research Center funded by the U.S. Department of Energy.

## Author contributions

C.W., J.Z. and L.H. conceived the idea and designed the experiments. C.W. carried out the aforementioned experiments. J.D. created the schematic representations herein. Y.Y. acquired the SEM images. L.X. and K.F. helped prepare the samples and analyse the data. C.W., K.F., S.D.L., J.Z. and L.H. analysed the data and wrote the paper. All authors contributed in revising the manuscript.

## Additional information

**Competing interests:** The authors declare no competing financial interests.

