## [Peer Review File · Nature Communications]

Reviewers' comments:

Reviewer #1 (Remarks to the Author):

This work presents a smart inverted-battery design. Unlike ordinary batteries where electrons are doing work in external circuits and ions are traveling within the batteries to complete the electrochemical reactions, in this "electron battery" design, the cathode and the anode are short-circuited and ions are released to the external circuit. The ions traveling in the external circuit can thus interact with biological systems and perform desirable functions. While the demonstrations of operating bio-systems with the electron batteries appear to be preliminary in my opinion, they have clearly proved the concept and will open new opportunities for using battery innovations for ion-controlled processes. Therefore, I suggest acceptance of the manuscript for publication after the authors provide more discussions or clarifications on the following questions:

1. In the demonstration shown in Fig. 4, is the motion of Cu ions caused by the electric field generated between the two electrodes, or by the motion of Li ions?
2. For the living cell experiment, how does the Na ion current trigger the Ca ion release? Is that a unique function of Na ions? Can it be stimulated by other ions or processes?

Reviewer #2 (Remarks to the Author):

The authors claimed an inverted battery in which ions travel through an external circuit while electrons are transported internally. The concept is novel but the design is also based on a sealed traditional battery as shown in Fig.4a, the whole inverted battery also depends on the electrochemical reactions of a traditional battery, so the novelty becomes doubtful. Further, the natural grass that has been used as an ionic cable in this work shows no obvious advantages compared to other kinds of materials, why use the grass? And also the results are not convinced enough, for example, some control experiments correspond to Fig. 4 are needed, how about the movement of copper ions without the battery? How did they measure the current? and how did they choose the anode, the cathode, the organic electrolyte, the exchange membrane? how about the results after changing some parameters of this battery? It is obviously that the claimed "inverted battery" needs more evidence to convince the readers.

Reviewers' comments:

Reviewer #1 (Remarks to the Author):

The authors have addressed my questions. I recommend acceptance of the MS.

Reviewer #2 (Remarks to the Author):

The authors have addressed most of my concerns, now I agree to accept it to be published in Nat. Commun.

Reviewer #3 (Remarks to the Author):

This paper develops a novel idea of inverted battery design in which ions instead of electrons, travel through the external circuit to interact with a bio-system. The authors also show its potential in biological applications. Although the design is in its initial stage, I believe this will interest researchers in both the battery and biology field. I recommend publication after they address the following questions.

1. In figure 2A left (traditional batteries), is it ionic system or electronic system in the external circuit? In traditional batteries, the electrons go through external electrical cables to interact with electronic devices.
2. Is there electrochemical reaction happening in the electron battery? According to the authors mentioned that 'no electrochemical reactions happen during the interaction'. Does it mean that there will be accumulation of electrons at the electrode-electrolyte interface during the operation? Please clarify
3. In Fig. 4. What happens after 10 mins? Was it still moving or it stabilized?
4. The use of grass stem is novel and environment friendly. However, for researchers who want to continue this topic, they might want some more standard or robust ion cables. Do the authors have any suggestions or standards?
5. In the biological application, what's the typical way of ions interacting with the bio-system? What's the advantage of the method reported here over the typical or old method?

REVIEWERS' COMMENTS:

Reviewer #3 (Remarks to the Author):

The paper is good for publication now.

Itemized list of response to the editor' and reviewer' remarks
(Black italic: Editor's or reviewer's remarks; Blue type: Our response)

Reviewer: 1

Comments:

This work presents a smart inverted-battery design. Unlike ordinary batteries where electrons are doing work in external circuits and ions are traveling within the batteries to complete the electrochemical reactions, in this “electron battery” design, the cathode and the anode are short-circuited and ions are released to the external circuit. The ions traveling in the external circuit can thus interact with biological systems and perform desirable functions. While the demonstrations of operating bio-systems with the electron batteries appear to be preliminary in my opinion, they have clearly proved the concept and will open new opportunities for using battery innovations for ion-controlled processes. Therefore, I suggest acceptance of the manuscript for publication after the authors provide more discussions or clarifications on the following questions:

Reply to the Reviewer:

We thank the reviewer's positive comments on the novelty of the work as “*a smart inverted-battery design*”. The reviewer also considers that “*they have clearly **proved the concept** and will **open new opportunities** for using battery innovations for ion-controlled processes*”.

1. In the demonstration shown in Fig. 4, is the motion of Cu ions caused by the electric field generated between the two electrodes, or by the motion of Li ions?

Reply to the Reviewer:

We thank the reviewer for the comments. The motion of Cu ions is caused by the electric field generated between the two electrodes due to the Li ion concentration difference. When the cathode and anode were shorted-circuited, Li ions enter the electrolyte from the anode and diffuse towards the cathode. During this process, the Li ion concentration difference between the two electrodes creates an electrical field that drives the Cu ions in the same direction as the movement of Li ions.

This discussion has been highlighted in the revised manuscript on page 15 and shown below.

When the electron battery is connected to the cotton string by two grass ionic cables, the lithium ions will diffuse from the anode to the cathode through the external ionic circuit to form an ionic current. The flow of Li^+ will create an electrical field, where the blue colored copper ions will drift towards the cathode.

2. For the living cell experiment, how does the Na ion current trigger the Ca ion release? Is that

a unique function of Na ions? Can it be stimulated by other ions or processes?

Reply to the Reviewer:

Thank you for the comments. The Ca²⁺ wave process of the living cell can be triggered by multiple stimuli, such as mechanical means as well as electrical fields.¹⁻³ This is not unique to the function of Na ions and is similar to the relationship between the Cu and Li ions. In this case, the Ca²⁺ wave was triggered by the electrical field formed by the Na ion current. Therefore, the other ions can also be used to form ionic current and trigger the Ca²⁺ wave. The Na ion is used in this work simply because it is part of the buffer solution and is not harmful to the living cells.

1. D'Andrea, P. & Vittur, F. Propagation of intercellular Ca²⁺ waves in mechanically stimulated articular chondrocytes. *FEBS Lett.* **400**, 58–64 (1997).
2. D'Hondt, C., Ponsaerts, R., Srinivas, S. P., Vereecke, J. & Himpens, B. Thrombin inhibits intercellular calcium wave propagation in corneal endothelial cells by modulation of hemichannels and gap junctions. *Investig. Ophthalmol. Vis. Sci.* **48**, 120–133 (2007).
3. Klepeis, V. E., Cornell-Bell, A. & Trinkaus-Randall, V. Growth factors but not gap junctions play a role in injury-induced Ca²⁺ waves in epithelial cells. *J. Cell Sci.* **114**, 4185–4195 (2001).

Reviewer #2 (Remarks to the Author):

The authors claimed an inverted battery in which ions travel through an external circuit while electrons are transported internally. The concept is novel but the design is also based on a sealed traditional battery as shown in Fig.4a, the whole inverted battery also depends on the electrochemical reactions of a traditional battery, so the novelty becomes doubtful.

Reply to the Reviewer:

We appreciate that the reviewer considers **the concept of the inverted battery to be novel**. It is true that both our inverted battery and traditional battery depend on electrochemical reactions. Traditional batteries have been used for more than a hundred years to convert electrochemical energy to electrical energy. To the best of our knowledge, there are numerous types of traditional batteries that have evolved over many generations however, they have never been used as an ion generator based on the proposed inverted battery design.

Although the inverted battery uses similar electrode materials with traditional battery, the device configurations (internal and external circuits, as Fig. R1 shows) are significantly different. For the traditional battery, the anode and cathode are separated by an ionic medium that allows ions to diffuse through while electrons travel via an external circuit. For the electron battery, an electrical medium is used to conduct electrons internally and the ions are transported through the external circuit.

Fig. R1. Schematics of a traditional battery (left) and an electron battery (right) interacting with an ionic system.

Their electrochemical behaviors are also different when they interact with ionic/bio-systems. The traditional battery exhibits an onset voltage (V_E) shown as an I-V curve, where the electrochemical reaction occurs (Fig. R2). The inverted battery exhibits different behavior (following Ohm's Law) where the current changes linearly with potential (Fig. R2).

Fig. R2. Illustration of I-V curves where a traditional battery and an electron battery interact with an ionic system. The curve of the electrical electrode exhibits an onset voltage (V_E), where the electrochemical reaction occurs. The ionic electrode exhibits different behavior (following Ohm's Law) where the current changes linearly with potential.

The functions of a traditional battery and our inverted battery are also completely different. The traditional battery is used as the electrical energy supply for electrical devices, while the inverted battery could be used as the ion source to cause interactions with ionic/bio-systems.

For potential applications, the inverted battery can be used for the treatment of nerve system damage, bipolar disorder (via a controllably released Li^+ source), heart disease (by altering contraction), Alzheimer's/Parkinson's disease (devices to control neuron stimulation), muscle stimulation as well as novel biosensors that can monitor human health.

Therefore, we believe our inverted battery concept is novel and will spur additional studies on ion-controlled processes for a range of applications, especially electro-biomedical applications.

Further, the natural grass that has been used as an ionic cable in this work shows no obvious advantages compared to other kinds of materials, why use the grass?

Reply to the Reviewer:

We thank the reviewer for the comments. The function of the ionic cables is to conduct ions. Most of the ionically conductive electrolytes are in liquid form. In order to create a cable-like conductor, mechanically robust materials, such as a cotton string or grass, are required to contain the liquid electrolyte. Note that a cotton string and natural grass are structurally different and thus, liquid electrolyte is stored in distinct ways. The soaked cotton string stores the electrolyte around the material (Figure R3A), while the natural grass possesses vertically aligned micro-channels that can hold the electrolyte (Figure R3B). Therefore, the natural grass retains the liquid electrolyte while maintaining flexibility, which is advantageous for ionic cables for the demonstration of the inverted battery concept in this work.

The statement has been added and highlighted in the revised manuscript on page 11 and shown below. The diagram below has also been added to the Supplementary Information.

In order to create a cable-like conductor, mechanically robust materials, such as a cotton string or grass, are required to contain the liquid electrolyte. Note that a cotton string and natural grass are structurally different and thus, liquid electrolyte is stored in distinct ways. The soaked cotton string stores the electrolyte around the material (Supplementary Figure 1A), while the natural grass possesses vertically aligned micro-channels that can hold the electrolyte (Supplementary Figure 1A-B). Therefore, the natural grass retains the liquid electrolyte while maintaining flexibility, which is advantageous for ionic cables.

Fig. R3. The illustration of ionic cables. (A) The cotton ionic cable is made of an electrolyte-soaked cotton string, where the electrolyte surrounds the cotton fiber. (B) In the grass ionic cable, the electrolyte is stored within the micro-aligned grass channels, which prevents electrolyte loss.

And also the results are not convinced enough, for example, some control experiments correspond to Fig. 4 are needed, how about the movement of copper ions without the battery?

Reply to the Reviewer:

We appreciate the reviewer for their constructive suggestion. To answer the reviewer's comment,

additional control experiments were conducted where copper ions diffused without the electron battery. According to Fig. R4, when no current was applied to the string, the blue copper ions randomly diffused along the string from the center towards the ends in both directions. This behavior is drastically different from the electron battery's driven process, where the ions mainly drift in one direction towards the cathode side. This further confirms that the proposed electron battery can cause ions to migrate in an ionic system.

Fig. R4. Digital images of the initial and diffused states for the control experiments. In the control experiments, no current was applied and the blue copper ions randomly diffused along the cotton string in both directions.

The results have been added into the new Fig. 4 and highlighted in the revised manuscript on pages 15 and 16.

In the control experiment, when no current was applied to the string, the blue ions randomly diffused along the string from the center (F) towards the ends in both directions (G). This behavior is drastically different from the electron battery's driven process.

Fig. 4. Characterization of the electron battery. (A) Schematic of the electron battery with a lithium metal anode, V_2O_5 nanowire cathode, and grass ionic cables. (B) Discharge profiles of the electron batteries controlled by ionic cables with different ionic resistances. The inset is the EIS spectra of the corresponding electron batteries. (C) A digital image of the electron battery demonstration to drive blue copper ions along a $LiNO_3$ solution soaked cotton string (~1 mm thick and 8 cm long). Digital images of (D) the initial and (E) moved state of blue copper ions before and after being driven by the electron battery. This indicates that the electron battery can cause ions to migrate in an ionic system. Digital images of (F) the initial and (G) diffused states for the control experiments. In the control experiments, no current was applied and the blue copper ions randomly diffused along the cotton string in both directions.

How did they measure the current?

Reply to the Reviewer:

Thank you for the comment. In the experiments, a Keithley 2400 SourceMeter was connected to the circuit to measure the current. This information is now added in the revised manuscript on page 22 and highlighted below.

In the experiments, a Keithley 2400 SourceMeter was connected to the circuit to measure the current.

How did they choose the anode, the cathode, the organic electrolyte, the exchange membrane?

Reply to the Reviewer:

We appreciate the reviewer for the concern. Any traditional battery materials (anode, cathode, organic electrolytes) have potential in our inverted battery design due to similarities in the electrochemical process.

For anode, lithium (Li) metal was selected as the anode in this work to achieve a high cell voltage since Li has the highest reduction potential (-3.04 V). As we mentioned in the manuscript, the electron battery has potential as the ion source to interact with bio-systems. For these specific applications, Li, Na, K, Ca, et al. can be selected as the anode materials to generate Li^+ , Na^+ , K^+ , Ca^{2+} , respectively. For example, lithium has been used as one of the most effective treatment methods for bipolar disorder. If an implantable electron battery with a lithium anode acted as the Li^+ source, lithium can be controllably released as needed, which could be a more effective and safer way to treat bipolar disorder.

For cathodes, they were chosen according to the specific anode material used. In this work, V_2O_5 nanowires acted as the cathode to enable fast Li^+ diffusion and high discharge rates. These properties are achieved due to the nanostructure of the V_2O_5 . Moreover, V_2O_5 starts from the non-lithiated state, therefore, no initial charge process was needed to use the electron battery. Note that other cathode materials used in traditional lithium ion batteries can also be employed here.

For electrolytes, they were selected based on the electrode materials. Here, the commonly used LiPF₆ in EC/DEC (1:1 by volume) electrolyte was used for the Li metal anode and its corresponding cathode material. Note that other Li^+ -based electrolytes can also be used with Li metal anodes. If Na, K, et al. were used as the anode material, the corresponding electrolytes would contain Na^+ or K^+ , respectively.

For ion exchange membranes, their main function is to prevent the organic electrolyte within the electrodes from mixing with the aqueous electrolytes within the ionic cables. Therefore, a membrane with low solvent permeability is preferred. A high ionic conductivity is also preferred to lower the working resistance. The ion exchange membranes consisted of two types: anion exchange membranes (AEMs) and cation exchange membrane (CEMs). The combination of

these membranes also results in different behavior. The diagram in Figure R5 shows that different membrane combinations cause drastic ion concentration changes. In some configurations, these sections act as an ionic pump to selectively change the local ion concentrations. In this work, two CEMs were used for both electrodes (Figure R5D), therefore, there is no local ion concentration change.

The above sections and the diagram shown below have been added to the Supplementary Information. Note that these additions are referred to on page 14 of the revised manuscript.

(Supplementary Figure 2; Supplementary Methods section for IEM selection rules).

Fig. R5. The change of ion concentration within different ion exchange membrane sections. (A) When an AEM and CEM are used as the cathode and anode, respectively, the ion concentration on the cathode ionic cable side will decrease, while the ion concentration in the ionic system will increase. (B) When the AEM and CEM are reversed, the ion concentration on the anode side of

the ionic cable will increase, while the ion concentration in the ionic system will decrease. (C) When an AEM is used for both electrodes, the ion concentration on the cathode side of the ionic cable will decrease, while the ion concentration on the anode side of the ionic cable will increase. (D) When both electrodes are CEMs, the ion concentration in each part stays constant.

How about the results after changing some parameters of this battery? It is obviously that the claimed "inverted battery" needs more evidence to convince the readers.

Reply to the Reviewer:

We thank the reviewer for the constructive suggestion. As the reviewer suggested, we conducted additional experiments by changing some parameters of the electron battery.

Specifically, Cu and Zn were used as the cathode and anode for the electron battery, while the electrolytes were also changed to the corresponding solutions, $\text{Cu}(\text{NO}_3)_2$ and $\text{Zn}(\text{NO}_3)_2$, respectively.

Moreover, the combination of ion exchange membranes for this electron battery were also changed, where the CEM and AEM ion exchange membranes are used on cathode and anode sides, respectively. In this case, the electron battery was used as the ionic pump for water desalination applications. According to the diagram shown in Fig. R6A, when the electron battery is applied to salt water, the specific combination of ion exchange membranes allows only the Na^+ and Cl^- within the salt water to be pumped towards the cathode and anode sides, respectively. Therefore, the salt water will be desalinated.

Fig. R6B records the current profile when Cu/Zn electron battery was applied to a 5 ml 100 mM NaCl aqueous solution. When NaCl is removed from the salt water, the conductivity of the salt water will continue to decrease, causing the discharge current to also decrease (Fig. R6B). Fig. R6C shows the EIS spectra of the NaCl solution at different concentrations, and the spectra of the salt water desalinated by the electron battery over ~70 h of operation. The intercepts of the EIS spectra on the real axis corresponds to the resistance of the NaCl solution. When resistance versus concentration is plotted in log-scale, the points form and fit well to a straight line, which is used as the calibration curve (Fig. R6D). According to the resistance of the desalinated sample, the remaining NaCl concentration corresponds to ~7.2 mM or 420 ppm. Note that the calculated concentration is below 500 ppm, which corresponds to fresh water by definition.

Therefore, electron batteries with new electrodes, electrolytes, and a combination of ion exchange membranes were successfully demonstrated as an ionic pump for water desalination applications. These results indicate that the electron battery significantly differs from traditional batteries and can interact with ionic systems in the way proposed in our manuscript.

Fig. R6. An electron battery made of a Cu cathode and Zn anode for water desalination applications. (A) Illustration of an electron battery for a water desalination system. (B) Current profile of the electron battery during the water desalination process over 70 h of operation. (C) EIS spectra and (D) the corresponding resistances of the NaCl solution at different concentrations.

Editors' comments:

When resubmitting your paper, we also ask that you ensure that your manuscript complies with our editorial policies. Specifically, please ensure that the following requirements are met, and any relevant checklists are completed and uploaded as a Related Manuscript file type with the revised article:

Characterization of chemical and biomolecular materials:

<http://www.nature.com/ncomms/journal-policies/editorial-publishing-policies#Characterization-materials>

Reply to the Editors:

We thank the Editors for the comments. The statements according to the editorial policies were added to the revised manuscript and highlighted below.

Data availability. The data that support the findings of this study are available from the corresponding author upon request.

Additional information

Supplementary Information accompanies this paper at <http://www.nature.com/naturecommunications>

Competing financial interests: The authors declare no competing financial interests.

The characterization of synthesized V_2O_5 nanowires was conducted and added in the supplementary Figure 3. The corresponding statements have also been added to the revised manuscript and highlighted below.

The diameters of the V_2O_5 nanowires are around several tens of nanometers, while their lengths are several micrometers (SEM image in Supplementary Figure 3).

Materials Characterization. The morphologies of the grass stem and V_2O_5 nanowires were conducted with a Hitachi SU-70 FEG-SEM at 10 kV.

Supplementary Figure 3. The SEM image of V₂O₅ nanowires synthesized by hydrothermal method.

Itemized list of response to the editor' and reviewer' remarks
(Black italic: Editor's or reviewer's remarks; Blue type: Our response)

Reviewers' comments:

Reviewer 1:

The authors have addressed my questions. I recommend acceptance of the MS.

Reply to the Reviewer:

We appreciate the reviewer's acceptance.

Reviewer 2:

The authors have addressed most of my concerns, now I agree to accept it to be published in Nat. Commun.

Reply to the Reviewer:

We appreciate the reviewer's acceptance.

Reviewer 3:

This paper develops a novel idea of inverted battery design in which ions instead of electrons, travel through the external circuit to interact with a bio-system. The authors also show its potential in biological applications. Although the design is in its initial stage, I believe this will interest researchers in both the battery and biology field. I recommend publication after they address the following questions.

Reply to the Reviewer:

We appreciate the reviewer's positive comments describing the work as "*a novel idea of inverted battery design*". We agree with the reviewer that "*this will interest researchers in both the battery and biology field*".

1. In figure 2A left (traditional batteries), is it ionic system or electronic system in the external circuit? In traditional batteries, the electrons go through external electrical cables to interact with electronic devices.

Reply to the Reviewer:

We thank the reviewer for the concern. Normally, the traditional batteries do interact with electronic devices. Fig. 2A is meant to compare the difference when both batteries interact with an ionic system, which therefore results in different electrochemical behavior and I-V curves as

shown in Fig. 2B. The statement was mentioned and highlighted in the manuscript on page 8 and is included below.

When these two types of batteries interact with an ionic system where the charge carriers are cations and anions, their electrochemical behaviors significantly differ.

2. Is there electrochemical reaction happening in the electron battery? According to the authors mentioned that ‘no electrochemical reactions happen during the interaction’. Does it mean that there will be accumulation of electrons at the electrode-electrolyte interface during the operation? Please clarify

Reply to the Reviewer:

We thank the reviewer for the comments. There is an electrochemical reaction happening in the electron battery. We are sorry for the confusion. The sentence has been changed in the revised manuscript on page 5 and highlighted below. When the electron battery interacts with an ionic system, the electrochemical reactions happen in the electrode of the electron battery so that the ions can be generated and conducted through ionic cables to interact with the ionic system (Fig. R1). Since the interaction is a process between ions, there is not electrochemical reaction happening in the ionic system, which is critical for bio-systems because electrochemical reactions can damage cells and tissues.

Fig. R1. Schematic of the electron battery interacting with an ionic system. The electrochemical reactions happen in the electrodes inside the battery, while the generated ions interact with the ionic system without causing electrochemical reactions.

“Moreover, since no electrochemical reactions happen in the bio-system during the interaction, the electron battery can directly apply electrochemical energy to the bio-systems without causing decomposition of the electrolyte and tissues.”

3. In Fig. 4. What happens after 10 mins? Was it still moving or it stabilized?

Reply to the Reviewer:

We thank the reviewer for the concern. For both the electron battery driven diffusion and the free diffusion, the experiments took about 10 mins to show any significant difference. After 10 mins, the ions driven by the electron battery still moved in the same direction, while the random diffusion ions spread on the cotton string until uniformly distributed.

4. The use of grass stem is novel and environment friendly. However, for researchers who want to continue this topic, they might want some more standard or robust ion cables. Do the authors have any suggestions or standards?

Reply to the Reviewer:

We appreciate the reviewer comment that **the use of grass stem is novel and environment friendly**. For ionic cables, the ionic conductivity, mechanical strength, and the flexibility are critical parameters. For biological applications, the biocompatibility of the ionic cable is also important. In our group, besides grass stems, we are also developing ionic cables with hydrogels, which will meet the standards listed above, including excellent biocompatibility.

5. In the biological application, what's the typical way of ions interacting with the bio-system? What's the advantage of the method reported here over the typical or old method?

Reply to the Reviewer:

We thank the reviewer for the query. As we have mentioned in the introduction of the manuscript on page 4 (highlighted in the revised manuscript and shown below), the typical method of ions interacting with the bio-system is achieved with the patch clamp technique. In this technique, an electrochemical reference electrode, typically Ag/AgCl electrode, is used as a bridge to enable the electronic system to measure the voltage and current in the bio-system, which is still an indirect way to interact with the bio-system. In our method, the electron battery can be an independent device to interact with the bio-system directly. The function of the patch clamp technique is mainly limited to measure the voltage and current of cells. Our electron batteries, as the ion generator, can generate multiple types of ions for different applications, such as Li ions for bipolar disorder treatment, Ca ions to interact with muscles, Na and K ions to interact with the nervous system, *etc.* Since the electron battery is intrinsically an energy storage device, it can also potentially supply energy to the bio-system, for example, as a pacemaker to help regulate the function of the cardiovascular system. Therefore, the electron battery is a novel concept, which will enable many promising applications and will be of interest to researchers in both the fields of batteries and biology.

Another method to achieve an interaction between ionic and electronic systems is the use of an electrochemical reference electrode as a bridge. Based on this method, the patch clamp technique was developed to study the ionic channels in cells by Erwin Neher and Bert Sakmann, who were awarded the Nobel Prize in Physiology or Medicine in 1991 for their work⁴⁻⁶.

Editors' comments:

At the same time, we ask that you ensure your manuscript complies with our editorial policies. Please ensure that the following requirements are met, and any relevant checklist is completed and uploaded as a Related Manuscript file type with the revised article.

Characterization of chemical and biomolecular materials:

<http://www.nature.com/ncomms/journal-policies/editorial-publishing-policies#Characterization-materials>

Furthermore, your manuscript should comply with our format requirements, which are summarized on the following checklist:

http://www.nature.com/article-assets/npg/ncomms/authors/ncomms_manuscript_checklist.pdf

Reply to the Editors:

We thank the Editors for the comments. The manuscript has been carefully revised to follow the format requirements. The abstract has been tailored to 150 words according to the format requirements and highlighted below. The manuscript checklist has also been uploaded.

In a lithium-ion battery, electrons leave one electrode and go through an external electronic circuit to power devices while ions transfer through internal ionic media to meet with electrons at the opposite electrode. Inspired by the lithium-ion battery chemistry, we envision a cell that can generate a current of ions instead of electrons, so that those ions can be used for potential applications in bio-systems. Based on this concept, we, for the first time, demonstrate an “electron battery” configuration in which ions travel through an external circuit to interact with a bio-system while electrons are transported internally. As a proof-of-concept, we demonstrate the application of the electron battery by stimulating a monolayer of cultured cells, which fluoresce a Ca^{2+} wave at a controlled ionic current. Electron batteries with the ability to generate a tunable ionic current pave the way towards precise ion-system control in a broad range of biological applications.

Itemized list of response to the editor' and reviewer' remarks
(Black italic: Editor's or reviewer's remarks; Blue type: Our response)

Reviewers' comments:

Reviewer #3 (Remarks to the Author): The paper is good for publication now.

Reply to the Reviewer:

We appreciate the reviewer's previous feedback and acceptance of the revised manuscript.